# Exposure to Nanoplastic Particles Enhances *Acinetobacter* Survival, Biofilm Formation, and Serum Resistance

**DOI:** 10.3390/nano12234222

**Published:** 2022-11-27

**Authors:** Mingfeng Tang, Guoying Ding, Xiaoyu Lu, Qian Huang, Huihui Du, Guosheng Xiao, Dayong Wang

**Affiliations:** 1College of Biology and Food Engineering, Chongqing Three Gorges University, Wanzhou 404100, China; 2Medical School, Southeast University, Nanjing 210009, China

**Keywords:** nanoplastic, *Acinetobacter*, interaction, virulence

## Abstract

The interaction between nanoplastics and bacteria remains still largely unclear. In this study, we determined the effect of nanopolystyrene particle (NP) on a bacterial pathogen of *Acinetobacter johnsonii* AC15. Scanning electron microscopy (SEM) analysis indicated the aggregation of NPs from 10 μg/L to 100 μg/L on surface of *A. johnsonii* AC15, suggesting that *A. johnsonii* AC15 acted as the vector for NPs. Exposure to 100–1000 μg/L NPs increased the growth and colony-forming unit (CFU) of *A. johnsonii* AC15. In addition, exposure to 100–1000 μg/L NPs enhanced the amount of formed biofilm of *A. johnsonii* AC15. Alterations in expressions of 3 survival-related (*zigA*, *basD*, and *zur*), 5 biofilm formation-related (*ompA*, *bap*, *adeG*, *csuC*, and *csuD*), and 3 serum resistance-related virulence genes (*lpxC*, *lpxL*, and *pbpG*) were observed after exposure to 1000 μg/L NPs. Moreover, both CFU and survival rate of *A. johnsonii* AC15 in normal human serum (NHS) were significantly increased by 1–1000 μg/L NPs, suggesting the enhancement in serum resistance of *Acinetobacter* pathogen by NPs. In the NHS, expressions of 3 survival-related (*zigA*, *basD*, and *zur*), 9 biofilm formation-related (*ompA*, *bap*, *adeF*, *adeG*, *csuA/B*, *csuC*, *csuD*, *csuE*, and *hlyD*), and 3 serum resistance-related virulence genes (*lpxC*, *lpxL*, and *pbpG*) were affected by 1000 μg/L NPs. Expressions of 1 survival-related (*zigA*), 5 biofilm formation-related (*bap*, *adeG*, *csuC*, *csuD*, and *csuE*), and 3 serum resistance-related virulence genes (*lpxC*, *lpxL*, and *pbpG*) were also altered by 10 μg/L NPs after the addition of NHS. Therefore, exposure to NPs in the range of μg/L has the potential to enhance bacterial virulence by increasing their growth, biofilm formation, and serum resistance.

## 1. Introduction

The challenge from global plastic pollution has received the increasing attention [1]. The emerging plastic pollution causes substantial consequences on both the organisms and the environment [2,3]. Weathering of the plastic by certain processes, such as biological process, results in debris formation with the forms of microplastic or even nanoplastic [4,5]. Both microplastics and nanoplastics have been detected in various ecosystems, such as marine and aquatic environments [6,7], which suggests the exposure risk of these materials to human and environmental organisms [8]. Meanwhile, the increasing experimental evidence has demonstrated the potential toxicity of microplastics and nanoplastics on different organisms [9,10,11,12,13].

Besides toxicity of microplastics and nanoplastic themselves, interactions between environmental pollutants and microplastic or nanoplastic have also been received the attentions. Microplastics and nanoplastics can act as vectors for environmental pollutants and even further enhance their toxicity on organisms [14,15]. Moreover, microplastics can further interact with and act as a vector for environmental microorganisms [16,17]. Among microorganisms in the environment, some are bacterial pathogens. Once the bacterial pathogens are available to organisms, they will affect the host health [18,19]. In addition, microplastic and nanoplastic at high concentration (such as those in the range of mg/L) caused bacterial growth inhibition, and the inhibition from nanoplastic exposure was more severe than that from microplastic exposure [20]. However, the effect of nanoplastics at low concentrations (such as those in the range of μg/L) on microorganisms remains largely unclear.

Three Gorges Reservoir (TGR) is a reservoir located upper stream of Yangtze River in China. In the TGR, plastic pollutions, including the microplastics, have been detected in both sediments and surface water [21,22]. AC15 is an *Acinetobacter johnsonii* strain isolated from the TGR region [23]. *A. johnsonii* is an opportunistic pathogen. Using *Caenorhabditis elegans* as a model host, it was proven that *A. johnsonii* AC15 is a bacterial pathogen [23]. In this study, we used nanopolystyrene particle (NP) as the example of nanoplastics to determine the effect of NPs at low doses (mainly in the range of μg/L) on *A. johnsonii* AC15. Our data demonstrated that exposure to NPs in the range of μg/L had the potential to enhance the virulence of bacterial pathogen by increasing both bacterial growth and biofilm formation. The findings highlight the risk of exposure to NPs in the range of μg/L in enhancing virulence of bacterial pathogens. In addition, in this study, we raised some molecular endpoints to assess this effect of NPs in the range of μg/L by identifying dysregulated virulence genes for *A. johnsonii* AC15.

## 2. Materials and Methods

### 2.1. Characterization of NPs

The 100 nm NPs were supplied by Huge Biotechnol. Co. (Wuhan, China). NP properties were evaluated by Zeta potential, dynamic light scattering (DLS), Fourier transform infrared spectroscopy (FTIR) spectrum (Avatar 370, Thermo Nicolet, Madison, WI, USA), and transmission electron microscopy (TEM). The dried NP sample was used for the analysis. Both size distribution and zeta potential of NPs were analyzed using Nano Zetasizer (Nano ZS90, Malvern Instrument, Malvern, UK). The morphology and size of NPs were further observed under the TEM (JEOL Ltd., Tokyo, Japan). FTIR analysis was performed to examine the different peaks of vibrations on NPs. To determine the interaction between NPs and *A. johnsonii* AC15, scanning electron microscopy (SEM) was further performed to analyze the aggregation of NPs (10–1000 μg/L) on surface of *A. johnsonii* AC15.

### 2.2. Acinetobacter Strain

AC15 is our isolated *Acinetobacter* strain in the TGR region [23]. *A. johnsonii* AC15 is normally stored in broth glycerol at −80 °C. To determine the location of *A. johnsonii* AC15 in systematic evolutionary tree, *A. johnsonii* AC15 DNA was extracted. The 16S rRNA was amplified in 25 μL polymerase chain reaction (PCR) reaction system using an upstream primer (515F: GTGCCAGCMGCCGCGGTAA) and downstream primer (806R: GGACTACHVGGGTWTCTAAT). Reaction conditions: pre-denaturation (94 °C, 5 min), denaturation (94 °C, 30 s), annealing (52 °C, 30 s), and extension (72 °C, 30 s). After 1% agarose gel electrophoresis, PCR product was used for the sequencing. Sequencing data was compared with the information of published sequences in GeneBank. With the aid of BLAST in NCBI, a systematic evolutionary tree, including AC15 was constructed (Appendix A). AC15 shows 99% homologous to *Acinetobacter Johnson*, and thus AC15 was identified as *Acinetobacter Johnson* (Appendix A).

### 2.3. Analysis of A. johnsonii AC15 Growth

Single colony of *A. johnsonii* AC15 was inoculated in 20 mL LB medium and incubated at 37 °C overnight. In 150 mL LB medium, NPs (0.1–1000 μg/L) and *A. johnsonii* AC15 (1.5 mL) were added, and AC15 was incubated at 37 °C for 24 h. During the incubation, optical density (OD) values of 595 nm were recorded each hour [24]. Meanwhile, colony-forming unit (CFU) of *A. johnsonii* AC15 was analyzed by counting colony number of *A. johnsonii* AC15/mL. The experiments were repeated three times.

### 2.4. Adsorption Curve of NP during Bacterial Growth

*A. johnsonii* AC15 single colonies were inoculated in 20 mL LB liquid medium under the conditions of 37 °C and 150 rpm shaking for 12 h. In 4 vials of 800 mL LB liquid medium, 4 experimental groups (control, NP (1000 μg/L), 1 mL *A. johnsonii* AC15 bacterial solution, and NP (1000 μg/L) + 1 mL *A. johnsonii* AC15 bacterial solution) were prepared and incubated at 37 °C for 24 h. During the incubation, OD values of 600 nm were recorded per hour. The turbidity of each experimental group was recorded every two hours. To measure turbidity, the supernatant was taken after centrifugation of 20 mL of bacterial solution at 5000 rpm for 3 min. Meanwhile, the CFU values of each experimental group were measured. The experiments were carried out three times.

### 2.5. Adsorption Curve of NP under the Condition of Non-Growth of Bacteria

Four bottles of 400 mL LB liquid medium were prepared and cooled down to room temperature. Two of them were added with 1000 μg/L NPs. *A. johnsonii* AC15 single colonies were inoculated in two vials of 400 mL LB liquid medium at 37 °C for 12 h. After the incubation, 5000 rpm centrifugation was performed for 3 min, and the supernatant was discarded. For one of the vials, 400 mL NPs (1000 μg/L) and 400 mL LB liquid medium were added and mixed well. Another one was only added with 400 mL LB liquid medium and mixed well. The vials were standing cultured at 4 °C for 12 h. During the incubation, OD values of 595 nm were recorded per hour. The turbidity of each experimental group was recorded every two hours. CFU values of each experimental group were measured. The experiments were carried out three times. SEM analysis was further performed to analyze the aggregation of NP (1000 μg/L) on the surface of *A. johnsonii* AC15 during the incubation at 4 °C.

### 2.6. Biofilm Formation of A. johnsonii AC15

The 96 well microtiter plate assay was used to evaluate biofilm formation [25]. In each well, 100 μL NPs (0.1–1000 μg/L) and 10 μL *A. johnsonii* AC15 were added. After incubation at 37 °C for 12 h, concentration of *A. johnsonii* AC15 solution was adjusted to OD600 value to 0.1. After the further incubation at 37 °C for 36 h, the solution was removed, and the wells were washed with PBS buffer and dried naturally. Methanol (100 μL) was added to each well to fix biofilm for 15 min. After that, biofilms of *A. johnsonii* AC15 at the bottom of each well were stained with 1% crystal violet (100 μL) for 5 min at room temperature. Again, 33% glacial acetic acid (100 μL) was added to dissolve crystal violet for 30 min at 37 °C. The optical density for samples in each well was detected at 595 nm. SEM was also performed to directly observe the biofilm formation of *A. johnsonii* AC15. The experiments were repeated three times.

### 2.7. Serum Resistance Analysis

The serum resistance test was performed as described [26]. NPs (0.1–1000 μg/L) and *A. johnsonii* AC15 (100 μL) were added in LB medium, and the final volume was 10 mL. After incubation at 37 °C for 12 h, concentration of *A. johnsonii* AC15 solution was adjusted to OD600 value to 0.5. After washing with PBS buffer 3 times, *A. johnsonii* AC15 was resuspended in 1 mL PBS buffer. *A. johnsonii* AC15 suspension (100 μL) was mixed with 300 μL normal human serum (NHS) and incubated at 37 °C for 3 h. After that, 100 μL serum of each sample was coated on surface of LB agar medium to incubate at 37 °C for 24 h. CFU was analyzed as described above. The survival rate of *A. johnsonii* AC15 was expressed as ratio between CFU of *A. johnsonii* AC15 suspension added with NHS and that of *A. johnsonii* AC15 suspension without NHS addition. The experiments were repeated three times.

### 2.8. Real-Time PCR (qRT-PCR)

Total RNAs of *A. johnsonii* AC15 were extracted with Trizol. The concentration and purity of obtained RNAs were determined with a spectrophotometer. Reverse transcription reactions for cDNA synthesis were performed using Mastercycler gradient PCR system. SYBR Green qRT-PCR master mix was used to detect transcriptional alterations of genes in real-time PCR systems. The internal reference gene is 16S rRNA. Three biological replicates were performed. Primer information is given in Appendix A.

### 2.9. Data Analysis

Software of SPSS V26.0 (SPSS Statistics, Version 26.0; IBM, Armonk, NY, USA) was used for the statistical testing. One-way analysis of variance (ANOVA) and multiple comparisons (LSD tests) were used to examine difference between groups. A *p*-value of <0.01 (**) was considered to be statistically significant.

## 3. Results and Discussion

### 3.1. Characterization of NPs

TEM analysis showed spherical morphology of NPs (Figure 1A). DLS analysis indicated that the NP size was 102.4 ± 5.8 nm, and the zeta potential of NPs was −9.245 ± 0.491 mV. Before the exposure, the NP suspensions were sonicated at 100 W and 40 kHz for 30 min, and no obvious aggregation was observed in the NP suspensions after the sonication. Based on FTIR analysis, C−H telescopic vibration absorption peak on benzene ring was observed at 3026.729 cm^−1^, and absorption peak of the C−H telescopic vibration of methyl and methylene was observed at 2915.76 cm^−1^ (Figure 1B). Three larger absorption peaks between 1620 and 1450 cm^−1^ were backbone vibration absorption peaks of aromatic hydrocarbons, indicating the presence of benzene rings (Figure 1B). Absorption peaks at 758.36 cm^−1^ and 700.51 cm^−1^ are single-substituted extra-surface bending vibration absorption peaks of benzene rings (Figure 1B).

### 3.2. Interaction between NPs and A. johnsonii AC15

Based on SEM analysis, the size of *A. johnsonii* AC15 was larger than the examined NPs (Figure 2). After the exposure, a large number of NPs were observed to be adsorbed and aggregated on the surface of *A. johnsonii* AC15 (Figure 2). Moreover, the aggregation of NPs at concentrations from 10 μg/L to 1000 μg/L on *A. johnsonii* AC15 was concentration−dependent (Figure 2).

### 3.3. Effect of NP Exposure on Growth of A. johnsonii AC15

After exposure to NPs (10–1000 μg/L), we did not observe the obvious alteration in morphology of *A. johnsonii* AC15 (Figure 2). We further examined the effects of NPs at different concentrations on *A. johnsonii* AC15 growth. Based on the growth curve analysis, exposure to 1000 μg/L NP significantly increased the growth of *A. johnsonii* AC15 from the time of 12-h (Figure 3A). In addition, exposure to 100 μg/L NP also moderately increased the *A. johnsonii* AC15 growth (Figure 3A). In contrast, exposure to 0.1–10 μg/L NP did not affect the growth curve of *A. johnsonii* AC15 (Figure 3A).

Based on the AC15 CFU analysis, at both 12-h and 24-h, exposure to 100–1000 μg/L NPs obviously increased the CFU of *A. johnsonii* AC15 (Figure 3B). Different from this, at both 12-h and 24-h, exposure to 0.1–10 μg/L NPs could not influence CFU of *A. johnsonii* AC15 (Figure 3B).

### 3.4. Adsorption Curve of NPs during Bacterial Growth

According to the growth curve analysis, 1000 μg/L NP could significantly increase the growth of *A. johnsonii* AC15 (Figure 4A). Meanwhile, we observed that exposure to 1000 μg/L NPs could significantly increase the CFU of *A. johnsonii* AC15 at 12 and 24 h (Figure 4B). To determine the reason for the interaction between *A. johnsonii* AC15 and NPs, the turbidity was analyzed to reflect the number of NPs adsorbed by *A. johnsonii* AC15. Addition with 1000 μg/L NPs resulted in the decline in turbidity of *A. johnsonii* AC15 solution during the growth (Figure 4C).

### 3.5. Adsorption Curve of NPs under the Condition of Non-Growth of Bacteria

Under the condition of standing culture at 4 °C for 12 h, addition of 1000 μg/L NPs did not have a significant effect on the growth and CFU of *A. johnsonii* AC15 (Appendix A). According to the turbidity analysis, after the addition of 1000 μg/L NPs, the turbidity remained unchanged during the incubation process (Appendix A). After addition of 1000 μg/L NP, we did not observe the noticeable accumulation of NPs on the surface of *A. johnsonii* AC15 (Appendix A).

The surface charge of the NPs in the LB liquid medium was further analyzed by zeta potential assay. The zeta potential of NP was −9.5 ± 0.56 mV.

### 3.6. Effect of NP Exposure on Biofilm Formation of A. johnsonii AC15

Biofilm forming is an important virulence factor for bacterial pathogen [25]. Based on crystal violet staining and SEM analysis, exposure to 0.1–10 μg/L NPs did not affect the amount of biofilm formation of *A. johnsonii* AC15 (Figure 5A–C). In contrast, after exposure to 100 and 1000 μg/L NPs, a significant increase in amount of biofilm formation for *A. johnsonii* AC15 was observed (Figure 5A–C).

### 3.7. Effect of NP Exposure on Serum Resistance of A. johnsonii AC15

Serum resistance is another important virulence factor for bacterial pathogens, including *Acinetobacter* [26]. After the addition of NHS, we found that both CFU and survival rate of *A. johnsonii* AC15 were not altered by exposure to 0.1 μg/L NPs (Figure 6A,B). In contrast, both CFU and survival rate of *A. johnsonii* AC15 in NHS were significantly increased by exposure to 1–1000 μg/L NPs (Figure 6A,B).

### 3.8. Effect of NP Exposure on Expression of Virulence Genes in A. johnsonii AC15

With the concern on *Acinetobacter* survival, biofilm formation, and serum resistance, we selected 17 virulence genes to determine the effects of NP exposure on *A. johnsonii* AC15 [27,28]. Exposure to 1000 μg/L NPs increased expression of 3 survival-related virulence genes (*zigA*, *basD*, and *zur*) (Figure 7A). Among biofilm formation-related virulence genes, expressions of *ompA*, *bap*, *adeG*, *csuC*, and *csuD* were increased by exposure to 1000 μg/L NPs (Figure 7B). Among serum resistance-related virulence genes, exposure to 1000 μg/L NPs increased *lpxC*, *lpxL*, and *pbpG* expressions (Figure 7C).

### 3.9. Effect of NP Exposure on Expression of Virulence Genes in A. johnsonii AC15 in NHS

We further used these 17 virulence genes to investigate the effect of NP exposure on survival, biofilm formation, and serum resistance of *A. johnsonii* AC15 after the addition of NHS. In the NHS, expressions of survival-related virulence genes (*zigA*, *basD*, and *zur*) were upregulated by exposure to 1000 μg/L NPs, and the *zigA* expression was also upregulated by exposure to 10 μg/L NPs (Appendix A). After the NHS addition, expressions of biofilm formation-related virulence genes (*ompA*, *bap*, *adeF*, *adeG*, *csuA/B*, *csuC*, *csuD*, *csuE*, and *hlyD*) were upregulated by exposure to 1000 μg/L NPs, and the *bap*, *adeG*, *csuC*, *csuD*, and *csuE* expressions were also upregulated by exposure to 10 μg/L NPs (Appendix A). In the NHS, expressions of serum resistance-related genes (*lpxC*, *lpxL*, and *pbpG*) were upregulated by exposure to 10 and 1000 μg/L NPs (Appendix A).

During the past several years, more and more attention has been paid to the potential risks of nanoplastic in the environment. Nanoplastics can be accumulated in environmental animals and may affect multiple organs of organisms through the circulatory system [29]. Exposure to nanoplastic can result in several aspects of toxicity, such as oxidative stress, neurotoxicity, reproductive toxicity, inflammatory response, and metabolic disorders [30,31,32,33,34]. The transgenerational toxicity on organisms can be further detected after nanoplastic exposure [35,36,37,38]. Nevertheless, compared with the toxicity of nanoplastics on environmental animals or plants, there are few reports on the effects of nanoplastics on microorganisms. Some reports have suggested the role of microplastics as vector of environmental microorganisms [16,17]. In this study, based on SEM analysis, we observed the aggregation of a large number of NPs on the surface of *A. johnsonii* AC15 (Figure 2), which suggested that the *A. johnsonii* AC15 acted as the vector of nanoplastic particles. This observation may be very different from the interaction between microplastics and microorganisms.

Previous studies have suggested that nanoplastics at high doses (such as those in the range of mg/L) could cause harmful effects on the physiological processes of bacteria. For example, exposure to 50 nm NPs (200 mg/L) inhibited the anaerobic digestion of sludge and affected the growth [39]. Similarly, the growth of marine bacterium *Halomonas alkaliphile* was inhibited and the ecological function was interrupted after microplastic exposure [20]. Additionally, the positively charged NPs could further cause lethal effects on yeast cells [40]. In this study, we further examined the effect of NPs at low doses (mainly in the range of μg/L) on bacteria. According to the growth curve and CFU analysis, 100–1000 μg/L NPs could promote the growth of *A. johnsonii* AC15 (Figure 3). The effect of NPs in the range of μg/L on bacteria was very different from that of high doses. That is, exposure to NPs in the range of μg/L may potentially enhance the risk of bacterial pathogen on environmental organisms by increasing bacterial growth. Therefore, more attention should be paid to the possible adverse effects of long-term and low-dose exposure to nanoplastics on bacterial pathogens in the environment.

To determine why a large number of NPs were accumulated on the surface of bacteria, the turbidity of NP in the supernatant of bacteria during growth and non-growth processes were analyzed. NPs caused the decline in turbidity of bacterial solution during the growth (Figure 4C). Different from this, after the NPs addition, the turbidity of bacterial solution during non-growth process remained unchanged (Appendix A). These observations suggested that the bacteria were more likely to adsorb NPs during the growth process. The zeta potential of NP in LB medium was −9.5± 0.56 mV. Previous study has shown that positively charged polystyrene nanoplastics (such as PS-NH_2_) could efficiently transport across cell membranes, while negatively charged PS (such as PS-COOH) and neutral PS had little or no effect in transport [41]. In this study, our data implied that the accumulation of NPs on surface of bacteria may be not due to electrostatic interactions.

The potential adverse effect of NPs in the range of μg/L on bacterial pathogens was also reflected by the increase in biofilm formation of *A. johnsonii* AC15. Biofilm is an important virulence factor of *Acinetobacter* infection, drug resistance, and escape from host immune response [41]. After formation of *Acinetobacter* biofilm, its ability to resist adverse factors, such as dryness and lack of nutrition, is significantly enhanced, and its survival ability and spread infection with hosts are enhanced [42]. Some studies have shown that microplastics can provide bacteria with a new adhesion matrix and colonize to form a biofilm [43]. Unlike microplastics, nanoplastics are smaller in size and do not allow bacteria to colonize them. However, based on our observations, nanoplastics could be adsorbed on the surface of bacteria (Figure 2). Meanwhile, both crystal violet staining and SEM analysis have indicated that the amount of biofilm formation of *A. johnsonii* AC15 was increased significantly by exposure to 100–1000 μg/L NPs (Figure 5). Therefore, exposure to NPs at low doses may further enhance the risk of bacterial pathogen by increasing biofilm formation.

Some studies have suggested that the ability to resist human serum killing is main survival method of *Acinetobacter* in the host [44,45]. According to serum resistance analysis, we observed that exposure to 1–1000 μg/L NPs could even significantly increase CFU and survival of *A. johnsonii* AC15 in the NHS (Figure 6). Therefore, besides behavior and effect on *Acinetobacter* growth in the environment, NPs also affected serum-resistant property of *A. johnsonii* AC15. In addition, the correlation between high biofilm formation activity and high resistance to NHS has been well demonstrated for bacterial pathogens [46]. That is, the increase in biofilm formation caused by NPs might contribute to the enhancement in serum-resistant property of *A. johnsonii* AC15. The 1 μg/L is a predicted environmental concentration for NPs [47]. Our data further implied the exposure risk of NPs at the predicted environmental concentration for environmental animals by affecting serum-resistant property of bacterial pathogens.

The observed effects of NP exposure on *Acinetobacter* survival, biofilm formation, and serum resistance were associated with the alteration in expressions of related virulence genes. *Acinetobacter* survival and proliferation in the environment and host is the premise of *Acinetobacter* pathogenesis. Along with the increase in survival of *A. johnsonii* AC15, expressions of 3 survival-related virulence genes (*zigA*, *basD*, and *zur*) were increased by 1000 μg/L NP (Figure 7A). Similarly, in the NHS, the expressions of these 3 survival-related virulence genes were increased by 1000 μg/L NP (Appendix A). More importantly, the *zigA* expression was also increased by 10 μg/L NP after NHS addition (Appendix A), which suggested that the addition of NHS may enhance the effect of NPs to increase the *Acinetobacter* survival. *zigA* gene governs the zinc uptake system [48]. *basD* gene controls the iron uptake system [49]. Besides the involvement in the absorption of iron under human host iron deficiency conditions, the product of *basD* is necessary for *Acinetobacter* to survive and cause apoptosis of human alveolar epithelial cells [50]. *zur* gene encodes a transcriptional regulator required for control of oxidative stress and zinc metabolism [51].

Biofilm formation plays an important role in *Acinetobacter* adhesion, infection, drug resistance gene transmission, drug resistance, and escape from host immune response [52]. Accompanied with the increase in biofilm formation, expressions of 5 biofilm formation-related virulence genes (*ompA*, *bap*, *adeG*, *csuC*, and *csuD*) were also increased by 1000 μg/L NPs (Figure 7B). Different from this, after the NHS addition, expressions of all the examined biofilm formation-related virulence genes were increased by 1000 μg/L NPs (Appendix A). In addition, 5 of them could be further increased by 10 μg/L NPs (Appendix A). These observations implied that, under the NHS addition condition, the role of NP exposure in increasing the *Acinetobacter* biofilm formation was also enhanced. *ompA* is essential for adhesion to epithelial cells [53]. *Bap* contributes to biofilm formation and adhesion to eukaryotic host cells [54]. The biofilm formation of *Acinetobacter* is related to the overexpression of AdeFGH efflux pump including both *adeF* and *adeG* [55]. Csu fimbriae (*csuA/B*, *csuC*, *csuD*, and *csuE*) play a role in the initial steps of biofilm formation by allowing bacterial cells to adhere to abiotic surfaces and initiating the formation of micro colonies before the biofilm structure is fully developed [56]. *hlyD* is involved in the protein secretion during biofilm formation [56].

Moreover, after the exposure to 1000 μg/L NP, we observed both the serum resistance and alteration in expressions of 3 serum resistance-related virulence genes (*lpxC*, *lpxL*, and *pbpG*) (Figure 6 and Figure 7C). In contrast, after the NHS addition, expressions of these 3-serum resistance-related virulence genes were increased by 10 and 1000 μg/L NPs (Appendix A). These observations suggested that the responses of serum resistance-related virulence genes to NPs were enhanced under the NHS addition condition. Among serum resistance-related genes, *lpxC* and *lpxL* are lipopolysaccharide (LPS) related genes, and *Acinetobacter* LPS have the function of evading host immune responses and triggering host inflammatory responses [57]. Inhibition of LpxC protects mice from resistant *Acinetobacter baumannii* by modulating inflammation and enhancing phagocytosis [58]. *pbpG* encodes putative low-molecular-mass penicillin-binding protein 7/8 (PBP-7/8) and contributes to the *Acinetobacter* survival [59].

## 4. Conclusions

Together, we examined the effect of nanoplastic exposure in the range of μg/L on *A. johnsonii* AC15. The obvious aggregation of NPs on the surface of *A. johnsonii* AC15 was observed, suggesting the role of *A. johnsonii* AC15 as the vector of NPs. Meanwhile, after exposure to 100–1000 μg/L NPs, we observed the increased growth and CFU of *A. johnsonii* AC15, the enhancement in the amount of biofilm formation of *A. johnsonii* AC15, and the enhanced serum resistance of *A. johnsonii* AC15. The observed effects of NP exposure on survival, biofilm formation, and serum resistance of *A. johnsonii* AC15 were associated with the alterations in expressions of survival-related (*zigA*, *basD*, and *zur*), 5 biofilm formation-related (*ompA*, *bap*, *adeG*, *csuC*, and *csuD*), and 3 serum resistance-related virulence genes (*lpxC*, *lpxL*, and *pbpG*). In addition, the responses of survival, biofilm formation, and serum resistance-related virulence genes to NPs were enhanced under the NHS addition condition. Therefore, our results demonstrated the potential of nanoplastics in the range of μg/L in enhancing the virulence of bacterial pathogens by increasing their survival, biofilm formation, and serum resistance in the environment. Further investigation on the interaction between NPs in the range of μg/L and certain pathogens in the real environment is needed to be further carried out in the future.

## Figures and Tables

**Figure 1 nanomaterials-12-04222-f001:**
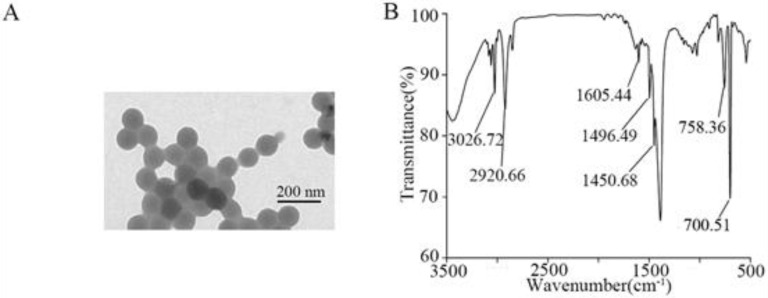
Nanopolystyrene particles (NPs) properties. (**A**) Transmission electron microscopy (TEM) image of NP suspension before the sonication. (**B**) FTIR spectrum of NPs.

**Figure 2 nanomaterials-12-04222-f002:**
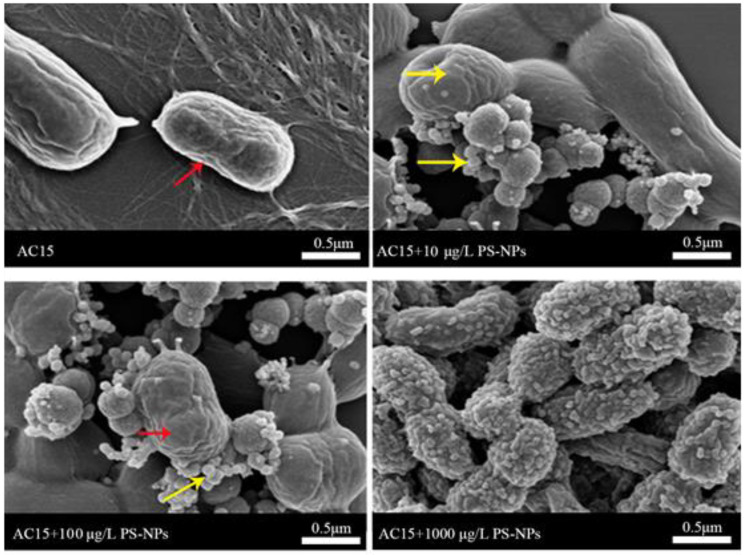
Scanning electron microscopy (SEM) images showing the aggregation of nanopolystyrene particle (NP) particles at concentrations of 10–1000 μg/L on surface of *A. johnsonii* AC15. Red arrowheads indicate *A. johnsonii* AC15. Yellow arrowheads indicate the aggregated NPs on surface of *A. johnsonii* AC15.

**Figure 3 nanomaterials-12-04222-f003:**
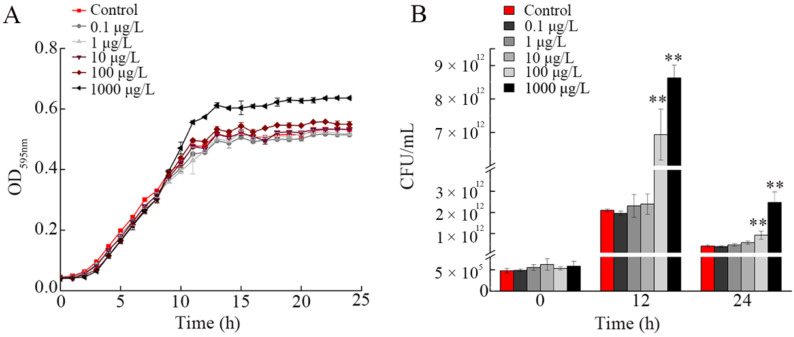
Effect of nanopolystyrene particle (NP) exposure on growth of *A. johnsonii* AC15. (**A**) Growth curves of *A. johnsonii* AC15 after exposure to NPs at different concentrations. (**B**) Effect of NP exposure on colony-forming unit (CFU) of *A. johnsonii* AC15 at different times. ** *p <* 0.01 vs. Control.

**Figure 4 nanomaterials-12-04222-f004:**
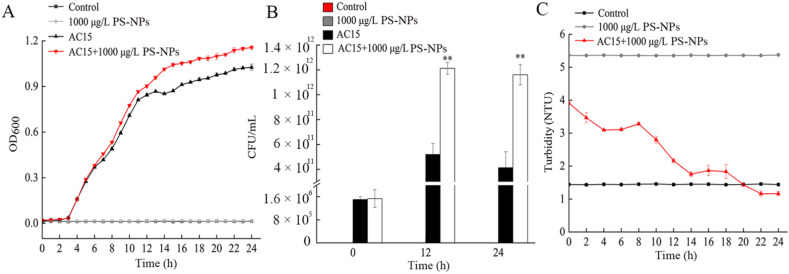
Adsorption curve of nanopolystyrene particles (NPs) during bacterial growth. (**A**) *A. johnsonii* AC15 growth curve after exposure to 1000 μg/L NP. (**B**) Effect of 1000 μg/L NP exposure on colony-forming unit (CFU) of *A. johnsonii* AC15 at different times. ** *p <* 0.01 vs. *A. johnsonii* AC15. (**C**) Turbidity analysis of bacterial solutions after cultivation at 37 °C for 24 h.

**Figure 5 nanomaterials-12-04222-f005:**
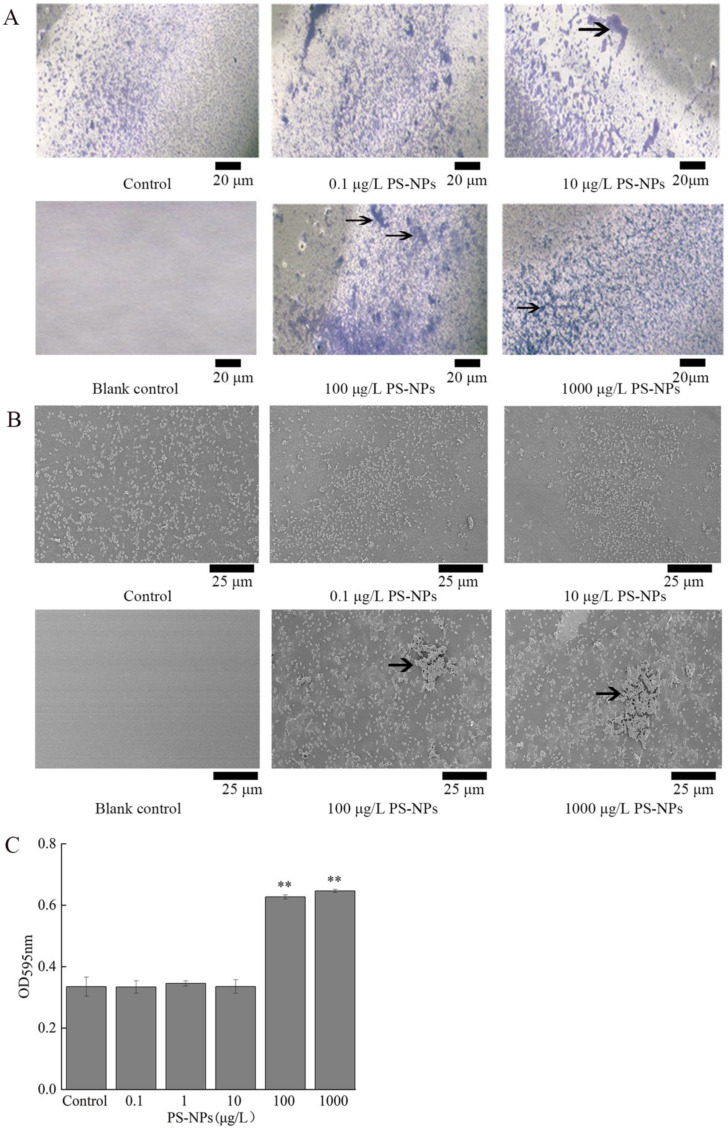
Effect of nanopolystyrene particle (NP) exposure at concentrations of 0.1–1000 μg/L on *A. johnsonii* AC15 biofilm formation. (**A**) Crystal violet staining images showing the effect of NP exposure at different concentrations on *A. johnsonii* AC15 biofilm formation. Arrowheads indicate the formed biofilm. (**B**) Scanning electron microscopy (SEM) images showing the effect of NP exposure at different concentrations on *A. johnsonii* AC15 biofilm formation. Arrowheads indicate the formed biofilm. (**C**) Effect of NP exposure at different concentrations on the amount of *A. johnsonii* AC15 biofilm formation based on the analysis of OD values of 595 nm. Blank control, without crystal violet staining. ** *p <* 0.01 vs. Control.

**Figure 6 nanomaterials-12-04222-f006:**
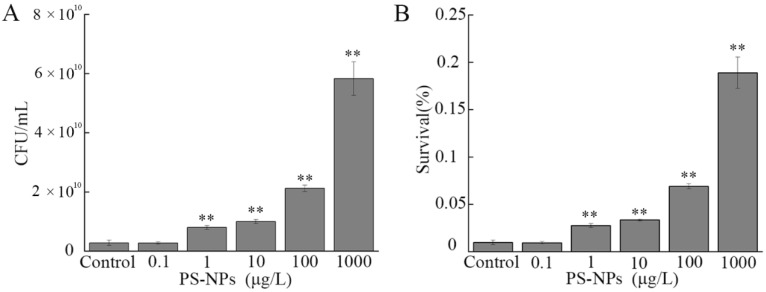
Effect of nanopolystyrene particle (NP) exposure on serum resistance of *A. johnsonii* AC15. (**A**) Colony-forming unit (CFU) of *A. johnsonii* AC15 in normal human serum (NHS) after exposure to NPs. (**B**) Survival rate of *A. johnsonii* AC15 in NHS after exposure to NPs. ** *p <* 0.01 vs. Control.

**Figure 7 nanomaterials-12-04222-f007:**
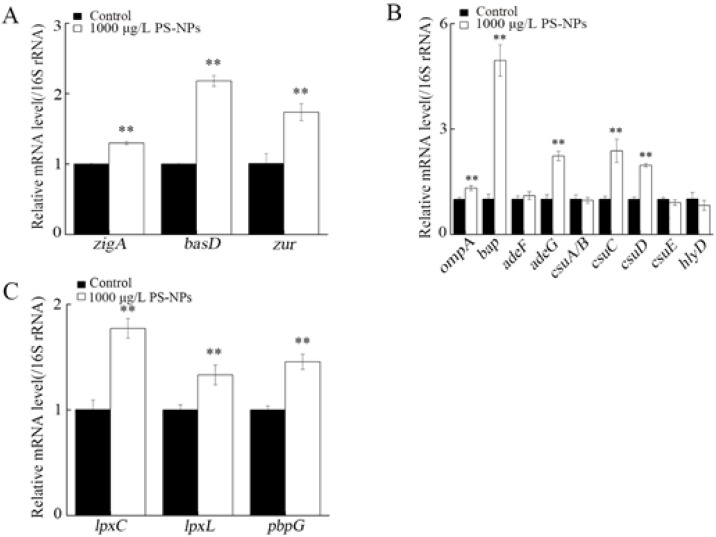
Effect of nanopolystyrene particle (NP) exposure on expression of virulence genes in *A. johnsonii* AC15. (**A**) Survival related genes. (**B**) Biofilm formation related genes. (**C**) Serum resistance related genes. ** *p <* 0.01 vs. Control.

## Data Availability

Not applicable.

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
