# Peer review of "Exposure to Nanoplastic Particles Enhances Acinetobacter Survival, Biofilm Formation, and Serum Resistance"

_nanomaterials, 2022, doi:10.3390/nano12234222_

Round 1

Reviewer 1 Report

·        When authors use the genus name of the bacteria itself, e.g. Acinetobacter then they should use sp. or spp., depending on what they mean, this applies to the entire manuscript including the title. What is the species of Acinetobacter? Section 2.2. Do you mean “johnsonii”? I can found it at https://www.ncbi.nlm.nih.gov/Taxonomy/Browser/wwwtax.cgi?name=Acinetobacter+johnsonii Correct please. So you probably should use whole name of your bacteria: Acinetobacter johnsonii AC15 throughout the whole manuscript.

·        Is. A. johnsonii a pathogen or rather opportunistic pathogen? Give more information about this to the Introduction.

·        Section 2.1. Give model of your devices, FTIR and microscope. Give more detailed description of the analyses. How were the samples prepared?

·        Section 2.6 – was the biofilm formation made on the polystyrene surface? Add the information.

·        Section 2.9. What software was used? Add the information.

·        In Figures captions all abbreviations should be defined.

·        Figure 1, 2 and 5 captions is not informative at all. Give information about the microscope and software.

·        Using AC15 in the whole text says nothing. Authors should use name of the bacteria A. johnsonii AC15, not only the strain symbol.

·        How many experiments were conducted in how many repeats? Add the information.

·        The graphs are technically undetailed.

·        On what basis were the nanoparticle concentrations chosen for the study?

·        The conclusions are written at a high level of generality. How did the nanoparticles cause an increase in the viability of the bacteria tested?

·        What is the application value of this research? Add the information.

·        What is the novelty of the study? Add the information.

Author Response

Reviewer 1:

When authors use the genus name of the bacteria itself, e.g. Acinetobacter then they should use sp. or spp., depending on what they mean, this applies to the entire manuscript including the title. What is the species of Acinetobacter? Section 2.2. Do you mean “johnsonii”? I can found it at https://www.ncbi.nlm.nih.gov/Taxonomy/Browser/wwwtax.cgi?name=Acinetobacter+johnsonii Correct please. So you probably should use whole name of your bacteria: Acinetobacter johnsonii AC15 throughout the whole manuscript.

Response: Thanks a lot for the comment from the reviewer. We have changed “Acinetobacter AC15” to “Acinetobacter johnsonii AC15” in “Abstract" and text.

  • Is. A. johnsoniia pathogen or rather opportunistic pathogen? Give more information about this to the Introduction.

Response: Thanks a lot for the comment from the reviewer. It is opportunistic pathogen, and we have explained this at the last paragraph in “Introduction”.

  • Section 2.1. Give model of your devices, FTIR and microscope. Give more detailed description of the analyses. How were the samples prepared?

Response: Thanks a lot for the comment from the reviewer. We have explained these further in “Materials and methods 2.1”:

“The 100 nm NPs were supplied by Huge Biotechnol. Co. (China). NP properties were evaluated by Zeta potential, dynamic light scattering (DLS), Fourier transform infrared spectroscopy (FTIR) spectrum (Avatar 370, Thermo Nicolet, USA), and transmission electron microscopy (TEM). The dried NP sample was used for the analysis. Both size distribution and zeta potential of NPs were analyzed using Nano Zetasizer (Nano ZS90, Malvern Instrument, UK). The morphology and size of NPs were further observed under the TEM (JEOL Ltd., Japan). FTIR analysis was performed to examine the different peaks of vibrations on NPs”.

  • Section 2.6 – was the biofilm formation made on the polystyrene surface? Add the information.

Response: Thanks a lot for the comment from the reviewer. We have explained this in “Materials and methods 2.6”:

“biofilms of A. johnsonii AC15 at the bottom of each well were stained with 1% crystal violet (100 μL) for 5 min at room temperature”.

  • Section 2.9. What software was used? Add the information.

Response: Thanks a lot for the comment from the reviewer. We have explained this in “Materials and methods 2.9:”

“Software of SPSS V26.0 (SPSS Statistics, Version 26.0; IBM, Armonk, USA) was used for the statistical testing”.

  • In Figures captions all abbreviations should be defined.

Response: Thanks a lot for the comment from the reviewer. We have all the related full names in the figure legends.

  • Figure 1, 2 and 5 captions is not informative at all. Give information about the microscope and software.

Response: Thanks a lot for the comment from the reviewer. We have provided more detailed information in figure legends for Figure 1, 2, and 5.

  • Using AC15 in the whole text says nothing. Authors should use name of the bacteria A. johnsoniiAC15, not only the strain symbol.

Response: Thanks a lot for the comment from the reviewer. We have changed “Acinetobacter AC15” to “Acinetobacter johnsonii AC15” in “Abstract" and text.

  • How many experiments were conducted in how many repeats? Add the information.

Response: Thanks a lot for the comment from the reviewer. Experiments were repeated for three times. We have indicated this in “Materials and methods 2.3, 2.4, 2.5, 2.6, 2.7, and 2.8”.

  • The graphs are technically undetailed.

Response: Thanks a lot for the comment from the reviewer. We have provided more information in revised Figure 5.

  • On what basis were the nanoparticle concentrations chosen for the study?

Response: Thanks a lot for the comment from the reviewer. We have explained this in the second paragraph in “Introduction”:

“In addition, microplastic and nanoplastic at high concentration (such as those in the range of mg/L) caused bacterial growth inhibition, and the inhibition from nanoplastic exposure was more severe than that from microplastic exposure [20]. However, the effect of nanoplastics at low concentrations (such as those in the range of μg/L) on microorganisms remains largely unclear”.

  • The conclusions are written at a high level of generality. How did the nanoparticles cause an increase in the viability of the bacteria tested?

Response: Thanks a lot for the comment from the reviewer. We have explained this in “Conclusion”:

“The observed effects of NP exposure on survival, biofilm formation, and serum resistance of A. johnsonii AC15 were associated with the alterations in expressions of survival-related (zigA, basD, and zur), 5 biofilm formation-related (ompA, bap, adeG, csuC, and csuD), and 3 serum resistance-related virulence genes (lpxC, lpxL, and pbpG)”.

  • What is the application value of this research? Add the information.

Response: Thanks a lot for the comment from the reviewer. We have explained this at the last paragraph in “Introduction”:

“The findings highlight the risk of exposure to NPs in the range of μg/L in enhancing virulence of bacterial pathogens. In addition, in this study, we raised some molecular endpoints to assess this effect of NPs in the range of μg/L by identifying dysregulated virulence genes for A. johnsonii AC15”.

  • What is the novelty of the study? Add the information.

Response: Thanks a lot for the comment from the reviewer. We have explained this at the last paragraph in “Introduction”:

“Our data demonstrated that exposure to NPs in the range of μg/L had the potential to enhance the virulence of bacterial pathogen by increasing both bacterial growth and biofilm formation”.

The revised MS is in the attachment.

Reviewer 2 Report

Drs. Xiao and Wang and their colleagues report on the interactions of nanoparticles (NP) of polystyrene and an environmental isolate of Acinetobacter both of which occur in the Three Gorges Reservoir where the study was conducted. The authors investigated the effects of NP on the bacteria and also tested the influence of NP on the expression of several virulence related genes. There are a number of questions related to the authors' conclusions that I would like to have addressed:

1) The authors cite their previous investigation in which they show that NP enhance the virulence of the Acinetobacter using a C. elegans bioassay. In reading this previous publication carefully, it is not apparent that the authors added NP to the E. coli strain normally used to maintain this culture and so that control is not evident. As this finding is key to the current study, such data should be presented and incorporated into this manuscript.

2) The authors employ LB as their test media. However, this media does not reflect the environment that the Acinetobacter is isolated in, so one might argue that the results and conclusions are a function of the laboratory growth conditions. It would be very useful for the authors to do some studies in growth conditions more relevant to the natural environment. This issue as well as the previous comment (number 1 above) need to be addressed.

3) The authors highlight several genes that may play a role, but do not discuss the potential implications of their results. One notable gene, bap was upregulated but there is very little discussed about this gene in the discussion. In reading another publication (Brossard and Campagnari, Infect. Immun. 80: 228) this gene is associated with cell surface hydrophobicity). Why were these specific genes targeted and what might be the significance of their findings?

4) In growth curves (Fig 3A) is absorption due to NP and media subtracted? Also from this reviewer's experience, microbial counts (CFU) are not always normally distributed, so log transforms are done for statistics. Did the authors test for normal distribution?

5) As a minor point, species names are lowercase.

6) In section 2.5, I would use the correct word incubation instead of "culturation". Also in an earlier sentence, please correct vails to vials.

7) What is the size distribution and chemical composition of the nanoparticles?

Author Response

Reviewer 2:

Drs. Xiao and Wang and their colleagues report on the interactions of nanoparticles (NP) of polystyrene and an environmental isolate of Acinetobacter both of which occur in the Three Gorges Reservoir where the study was conducted. The authors investigated the effects of NP on the bacteria and also tested the influence of NP on the expression of several virulence related genes. There are a number of questions related to the authors' conclusions that I would like to have addressed:

1) The authors cite their previous investigation in which they show that NP enhance the virulence of the Acinetobacter using a C. elegans bioassay. In reading this previous publication carefully, it is not apparent that the authors added NP to the E. coli strain normally used to maintain this culture and so that control is not evident. As this finding is key to the current study, such data should be presented and incorporated into this manuscript.

Response: Thanks a lot for the comment from the reviewer. In the last paragraph, what we stated was “Using Caenorhabditis elegans as a model host, it was proven that A. johnsonii AC15 is a bacterial pathogen”. This only means that infection with A. johnsonii AC15 could cause toxic effects in nematodes. In the previous study, we did not determine the effects of NPs.

2) The authors employ LB as their test media. However, this media does not reflect the environment that the Acinetobacter is isolated in, so one might argue that the results and conclusions are a function of the laboratory growth conditions. It would be very useful for the authors to do some studies in growth conditions more relevant to the natural environment. This issue as well as the previous comment (number 1 above) need to be addressed.

Response: Thanks a lot for the comment from the reviewer. The aim of this study was to determine the interaction between nanoplastic and A. johnsonii AC15. During this process, the LB was used for normal AC15 culturation. Introduction of other environmental factors will make us to meet the difficulty to make the exact conclusion on the interaction between nanoplastic and A. johnsonii AC15. Meanwhile, this suggestion is really a candidate direction for the future work based on the conclusion obtained in this study. We discussed this in “Conclusion”:

“The further investigation on the interaction between NPs in the range of μg/L and certain pathogens in the real environment is needed to be further carried out in the future”.

3) The authors highlight several genes that may play a role, but do not discuss the potential implications of their results. One notable gene, bap was upregulated but there is very little discussed about this gene in the discussion. In reading another publication (Brossard and Campagnari, Infect. Immun. 80: 228) this gene is associated with cell surface hydrophobicity). Why were these specific genes targeted and what might be the significance of their findings?

Response: Thanks a lot for the comment from the reviewer. We have explained the related genes dysregulated in “Discussion” one by one. Such as,

Bap contributes to biofilm formation and adhesion to eukaryotic host cells [44]. The biofilm formation of Acinetobacter is related to the overexpression of AdeFGH efflux pump including both adeF and adeG [45]. Csu fimbriae (csuA/B, csuC, csuD, and csuE) play a role in the initial steps of biofilm formation by allowing bacterial cells to adhere to abiotic surfaces and initiating the formation of micro colonies before the biofilm structure is fully developed [46]. hlyD is involved in the protein secretion during biofilm formation [47]”.

For the Bap, it is an important biofilm related gene. Besides the cited reference, the other below references further support this. Nevertheless, we do not exclude that this gene is also involved other biological processes.

-Tiwari V, Patel V, Tiwari M. In-silico screening and experimental validation reveal L-Adrenaline as anti-biofilm molecule against biofilm-associated protein (Bap) producing Acinetobacter baumannii. Int J Biol Macromol. 2018 Feb;107(Pt A):1242-1252.

-Fallah A, Rezaee MA, Hasani A, Barhaghi MHS, Kafil HS. Frequency of bap and cpaA virulence genes in drug resistant clinical isolates of Acinetobacter baumannii and their role in biofilm formation. Iran J Basic Med Sci. 2017 Aug;20(8):849-855.

4) In growth curves (Fig 3A) is absorption due to NP and media subtracted? Also from this reviewer's experience, microbial counts (CFU) are not always normally distributed, so log transforms are done for statistics. Did the authors test for normal distribution?

Response: Thanks a lot for the comment from the reviewer. Just as indicate by the reviewer, considering that the CFUs are not always normally distributed, we did not test for the normal distribution.

5) As a minor point, species names are lowercase.

Response: Thanks a lot for the comment from the reviewer. We have corrected this in the full text.

6) In section 2.5, I would use the correct word incubation instead of "culturation". Also in an earlier sentence, please correct vails to vials.

Response: Thanks a lot for the comment from the reviewer. We have corrected these in “Materials and methods 2.5” according to the comment from the reviewer.

7) What is the size distribution and chemical composition of the nanoparticles?

Response: Thanks a lot for the comment from the reviewer. We have explained the size distribution in “Results and Discussion 3.1”:

“DLS analysis indicated that the NP size was 102.4 ± 5.8 nm, and the zeta potential of NPs was -9.245 ± 0.491 mV. Before the exposure, the NP suspensions were sonicated at 100 W and 40 kHz for 30 min, and no obvious aggregation was observed in the NP suspensions after the sonication”.

In addition, the used NP is a commercial source, and the detailed chemical composition of the nanoparticles is not clear.

Please check the revised MS from the attachment.

Round 2

Reviewer 1 Report

I have no more comments.

Author Response

Thanks a lot for the comment from the reviewer.

Reviewer 2 Report

I had two notable concerns in the original submission. The first was the indication that nanoparticles (NP) were associated with decreased survival in a previous report using C. elegans as an assay. The authors have clarified this issue. The second point is the growth media (LB) does not reflect the natural environment and so the findings may be due to the physiological action of NP with Acinetobacter johnsonii in LB and not a general finding. This point is under investigation in future studies. The only remaining issue that I would like the authors to make is to put a bit more descriptive figure legend into Figure S1 (i.e., what type of phylogenetic analysis is being used)

Author Response

Comment: I had two notable concerns in the original submission. The first was the indication that nanoparticles (NP) were associated with decreased survival in a previous report using C. elegans as an assay. The authors have clarified this issue. The second point is the growth media (LB) does not reflect the natural environment and so the findings may be due to the physiological action of NP with Acinetobacter johnsonii in LB and not a general finding. This point is under investigation in future studies. The only remaining issue that I would like the authors to make is to put a bit more descriptive figure legend into Figure S1 (i.e., what type of phylogenetic analysis is being used).

Response: Thanks a lot for the comment from the reviewer. We have added more information in figure legend for Figure S1:

Figure S1. Systematic evolutionary tree including AC15. The form of labeled tree was used. After the sequencing, the data was compared with the published sequences in GeneBank. Using BLAST in NCBI website, the systematic evolutionary tree was constructed. In this evolutionary tree, AC15 was identified as Acinetobacter Johnson”.